# Distribution, diversity and persistence of *Listeria monocytogenes* in swine slaughterhouses and their association with food and human listeriosis strains

**Tamazight Cherifi**[1,2,3]*, **Julie Arsenault**[2,4], **Franco Pagotto**[5], **Sylvain Quessy**[1,3], **Jean-Charles Côté**[1,3], **Kersti Neira**[1,2,3], **Sylvain Fournaise**[6], **Sadjia Bekal**[7], **Philippe Fravalo**[1,2,4]*

1 Chaire de recherche en salubrité des viandes, Faculté de médecine vétérinaire, Université de Montréal, Saint-Hyacinthe, Quebec, Canada, 2 Centre de recherche en infectiologie porcine et avicole (CRIPA-FQRNT), Faculté de médecine vétérinaire, Université de Montréal, Saint-Hyacinthe, Quebec, Canada, 3 Groupe de recherche et d'enseignement en salubrité des aliments (GRESA), Faculté de médecine vétérinaire, Université de Montréal, Saint-Hyacinthe, Quebec, Canada, 4 Epidemiology of Zoonoses and Public Health Research Unit (GREZOSP), Faculté de médecine vétérinaire, Université de Montréal, Saint-Hyacinthe, Quebec, Canada, 5 Listeriosis Reference Service, Microbiology Research Division, Bureau of Microbial Hazards, Food Directorate, Health Products and Food Branch, Health Canada, Ottawa, Ontario, Canada, 6 Olymel, Boucherville, Quebec, Canada, 7 Laboratoire de santé publique du Québec, Sainte-Anne-de-Bellevue, Quebec, Canada

* tamazight.cherifi@umontreal.ca (TC); philippe.fravalo@lecnam.net (PF)

**Data Availability Statement:** All relevant data are within the paper and its Supporting Information files.

## Abstract

*Listeria monocytogenes* is the etiological agent of listeriosis, a major foodborne disease and an important public health concern. Contamination of meat with *L. monocytogenes* occurs frequently at the slaughterhouse. Our aims were; 1) to investigate the distribution of *L. monocytogenes* in the processing areas of four swine slaughterhouses; 2) to describe the diversity of *L. monocytogenes* strains by pulsed-field gel electrophoresis; 3) to identify persistent *L. monocytogenes* strains and describe their distribution; 4) to investigate the associations between persistence of strains and their following characteristics: detection in food isolates, detection in human clinical isolates, and the presence of benzalkonium chloride (BAC) resistance genes. Various operation areas within the four swine slaughterhouses were sampled on four occasions. A total of 2496 samples were analyzed, and *L. monocytogenes* was successfully isolated from 243 samples. The proportion of positive samples ranged from 32 to 58% in each slaughterhouse and from 24 to 68% in each operation area. Fifty-eight different pulsotypes were identified and eight pulsotypes, present in samples collected during 4 visits, were considered persistent. The persistent pulsotypes were significantly more likely to be detected in food ($P < 0.01$, exact $\chi^2$) and human clinical cases ($P < 0.01$, exact $\chi^2$), respectively. Among pulsotypes harboring the BAC *bcrABC* resistance cassette or the *emrE* multidrug transporter gene, 42.8% were persistent compared to 4.5% for pulsotypes without these resistance genes ($P < 0.01$, exact $\chi^2$). Our study highlights the importance of persistent *L. monocytogenes* strains in the environmental contamination of slaughterhouses, which may lead to repeated contamination of meat products. It also

**Funding:** This work was funded by the Natural Sciences and Engineering Research Council of Canada (NSERC) grant number RDCPJ 520873-17 and by industrial partners of the Chaire de recherche en salubrité des viandes to Philippe Fravalo -https://www.nserc-crsng.gc.ca/index_eng.asp -olymel.ca/en/?in The funders had no role in study design, data collection and analysis, decision to publish, or preparation of the manuscript.

**Competing interests:** Authors declare no competing of interest. Dr Sylvain Fournaise (from Olymel) contributed to the sampling design at the slaughterhouses and the interpretation of results, but had no role in laboratory analyses, statistical analyses or in reporting the results presented in this work. This does not alter our adherence to PLOS ONE policies on sharing data and materials.

shows that the presence of disinfectants resistance genes is an important contributing factor.

## Introduction

*Listeria monocytogenes* is a ubiquitous Gram-positive, rod-shaped, psychrophilic, food-borne bacterial pathogen. It is the etiologic agent of listeriosis, an infection characterized by fever, muscle aches, nausea and diarrhea [1]. More severe cases can lead to sepsis, meningitis, encephalitis and even death [2]. *L. monocytogenes* can contaminate raw fruits and vegetables, animal meat, unpasteurized dairy products and processed foods. It is a major concern for both the food industry and public health [3].

The environment of swine slaughterhouses can become contaminated with *L. monocytogenes*, which can then serve as a source of carcass contamination throughout the processing areas [4–8]. A slaughterhouse can be divided into eight operation areas: lairage, stunning, slaughtering and bleeding, dehairing, evisceration, chilling and hanging, cutting and deboning, and freezing and delivery [9], and each can become contaminated. Despite the cleaning and sanitary treatment, some *L. monocytogenes* strains can persist in the slaughterhouse and persistent strains have been shown to be responsible for repeated food contaminations [10–13]. In USA, a year 2000 multi-state listeriosis outbreak was linked to a persistent *L. monocytogenes* strain first identified in a 1988 listeriosis case [11]. One of the reasons for the occurrence of persistence in *L. monocytogenes* strains can be related to their resistance to industrial disinfectants [14]. Several studies revealed resistance to quaternary ammonium compounds (QACs) in some *L. monocytogenes* strains isolated from different food processing plants [15–17]. This resistance may have been caused by the presence of sublethal concentrations of the disinfectants due to the dilution effect that occurs during the industrial sanitation procedures [14]. Benzalkonium chloride (BAC) is a type of QAC widely used in slaughterhouses as a broad-spectrum hard surface disinfectant [18]. *L. monocytogenes* persistent strains often harbor adaptation and resistance genes to BAC, including the *bcrABC* resistance cassette [19], and *emrE*, a multidrug transporter gene [20].

To the best of our knowledge, in Canada, no study has been carried on the distribution, diversity and persistence of *L. monocytogenes* in swine slaughterhouses. Consequently, no information is available on the potential association between the presence of putative *L. monocytogenes* persistent strains in swine slaughterhouses and human listeriosis cases in Canada.

Our objectives were: 1) to investigate the distribution of *L. monocytogenes* in the processing areas of four swine slaughterhouses; 2) to describe the diversity of *L. monocytogenes* strains by pulsed-field gel electrophoresis; 3) to identify persistent *L. monocytogenes* strains and describe their distribution; 4) to investigate the associations between persistence of strains and their following characteristics: detection in food isolates, detection in human clinical isolates, presence of BAC resistance genes.

## Materials and methods

### Sampling procedures

Four swine slaughterhouses, labeled A, B, C and D, which cover 60% (from 20000 to 37500 swine per week) of pork meat production in Quebec, Canada, were sampled four times with a four- to five-month interval between visits, over a 16-month period, from October 2013 to February 2014. All samples were collected at least 12h after in-site cleaning and sanitation,

from six different operation areas within the slaughterhouses: 1) lairage, 2) slaughtering and bleeding, 3) dehairing, 4) evisceration, 5) chilling and hanging and 6) cutting and deboning. For each operation area, the surface of different sites, machines and materials, whether in regular contact with carcasses or meat or not, were sampled and are listed in S1 Table. Samples from two operation areas, dehairing and evisceration, were grouped together for later analyses purposes. Depending on the sites, machines and materials, the sampling surface varied from 5 cm$^2$ for knives to 1 m$^2$ for conveyor belts, doors, walls and floors. Before swabbing, each sampling surface was brushed vigorously using a sterile toothbrush to resuspend any possible residues. Fisherbrand lab wipes (Fisher Scientific, Ottawa, ON, Canada), pre-soaked in a D/E (Dey-Engley) neutralizing broth (Innovation Diagnostic, Saint-Eustache, QC, Canada) to neutralize a broad spectrum of disinfectants and antiseptics, were used to swab all sampling surfaces. Following swabbing, each toothbrush and lab wipe pair was transferred into a Nasco Whirl-Pak bag (Fisher Scientific), kept in a cooler with ice packs, and brought to the lab.

## Isolation of *L. monocytogenes*

First, 100 ml of a *Listeria* enrichment broth (University of Vermont Medium I; UVM-1; Innovation Diagnostics) was added into each Nasco Whirl-Pak bag containing a toothbrush and the lab wipe, mixed using a Seward Stomacher 400C Lab Blender (Cole-Parmer, Montreal, QC, Canada) for 1 min, and incubated for 48 hr at 30$^{\circ}$C. *L. monocytogenes* were isolated according to Pagotto et al [21]. Briefly, after 48 hr, positive UVM-1cultures were streaked on selective COMPASS *Listeria* agar (Innovation Diagnostics), and 10 mL of a second enrichment broth (modified Fraser broth; MFB; Innovation Diagnostics) were inoculated with 100 μL of negative cultures of UVM-1 primary broth and incubated at 35°C for 24 hr. Positive MFB cultures were streaked on selective COMPASS *Listeria* agar (Innovation Diagnostics) and incubated at 37°C for 24h.

Confirmation of *L. monocytogenes* isolates was done by testing for beta-hemolysis on sheep blood agar (Oxoid, Nepean, ON, Canada) and carbohydrate metabolism assays using xylose, rhamnose and mannitol. Two colonies per sample were randomly picked from sheep blood agar plates and the *L. monocytogenes* species was confirmed by PCR by the detection of *prf*A, *prs*, *lmo0737*, *lmo1118*, *orf2819*, *orf2110*, and *fla*A genes according to Kérouanton, Marault [22].

## Characterisation of *L. monocytogenes* isolates

**Serotyping.**  The obtained isolates were serotyped using the *L. monocytogenes* antisera for both O- and H-antigens (Denken Seiken Co., Ltd., Tokyo, Japan) according to the manufacturer's instructions.

**Ribotyping.**  Ribotyping was performed on all isolates with restriction enzyme EcoRI using the RiboPrinter microbial characterization system (DuPont Qualicon, Wilmington, DE, USA), according to the manufacturer's instructions and as previously described [23].

**Pulsed-Field Gel Electrophoresis (PFGE) typing.**  One isolate per sample and per ribotype was selected for PFGE typing (i.e. if two isolates from a same sample had different ribotype, both were selected for PFGE typing; otherwise, one isolate was randomly selected). In addition, for a subset of randomly selected samples with two isolates sharing the same ribotype, the two isolates were typed. PFGE typing was performed according to the Center for Disease Control and Prevention (CDC) PulseNet standardized protocol for *L. monocytogenes* typing [24] using the clamped homogeneous electric fields (CHEF-DR III, Bio-Rad Laboratories, Hercules, CA, USA). DNA was cleaved with ApaI and AscI, (New England Biolabs, Whitby, ON, Canada). *Salmonella* Braenderup was used as a reference bacterial strain and size

marker. The PFGE pulsotypes were compared against the PulseNet Canada database, housed at the National Microbiology Laboratory (NML, Winnipeg, MB, Canada).

### Distribution of *L. monocytogenes*-positive samples

The proportion of *L. monocytogenes*-positive samples was described according to the seasons, slaughterhouses, operation areas, contact of surfaces with carcasses or meat, and type of surface. A multi-level logistic regression was used to model the probability of detecting *L. monocytogenes* according to these five variables. The visit was included as a random effect. The Laplace method was used for estimation. From the full model, a backward manual procedure was used to select the final model, using a $P > 0.05$ as a criterion for rejection. Odds ratios were used to present results, with *P*-values adjusted for multiples comparisons using Tukey-Cramer tests. Analyses were performed using the GLIMMIX procedure of SAS v 9.4 (SAS Institute Inc., Cary, NC, USA).

### *L. monocytogenes* pulsotype diversity

A PFGE dendrogram was created using a combination of the PFGE fingerprints of both enzymes with BioNumerics version 6.06 (Applied Maths, Inc, Austin, TX, USA) based on the Unweighted pair group method with arithmetic mean (UPGMA) analysis of the Dice similarity coefficient at 1%. The diversity of pulsotypes according to the seasons, slaughterhouses and operation areas was described using rarefaction curves using iNEXT package [25] available on R v 3.5.2. To avoid bias due to the method of selection, only one strain per positive sample was randomly selected.

### Distribution of *L. monocytogenes* persistent pulsotypes

A persistent *L. monocytogenes* pulsotype was defined as a strain (based on PFGE pulsotype) isolated at least three times in the same slaughterhouse over the 16-month sampling period, with a four- to five-month interval between each sampling visit, in accordance with Keto-Timonen, Tolvanen [26] and Ortiz, López [27]. A multilevel logistic regression was used to model the contamination (yes/no) of samples with a persistent pulsotype of *L. monocytogenes* according to the slaughterhouses and operation areas, as described above for the analysis of *L. monocytogenes*-positive samples. This analysis was limited to *L. monocytogenes* positive samples, with one randomly selected isolate per sample.

### Association between *L. monocytogenes* persistent pulsotypes in the swine slaughterhouses and their detection in food and human clinical isolates of *L. monocytogenes*

Each *L. monocytogenes* pulsotype was classified either as persistent or not persistent based on criteria described above. They were compared with pulsotypes of *L. monocytogenes* isolates from food and human listeriosis cases in Quebec between the years 2000 and 2016 according to the database at the *Laboratoire de santé publique du Québec* (LSPQ; Sainte-Anne-de-Bellevue, QC, Canada). The association between persistence (persistent vs. non-persistent) of the pulsotype and detection in food or human clinical cases, respectively, was evaluated using exact $\chi^2$. The PFGE pulsotype was used at the unit of analysis.

### PCR amplification of the benzalkonium chloride (BAC) *bcrABC* resistance cassette and the *emrE* multidrug transporter gene from *L. monocytogenes* isolates

The presence of the benzalkonium chloride (BAC) *bcrABC* resistance cassette and the *emrE* multidrug transporter gene were screened by PCR on one randomly selected isolate per pulso-type. Briefly, colonies from selected isolates were transferred into a 1.5 ml microcentrifuge tube (FisherScientific), containing 50 μl of Chelex 100 resin (Bio-Rad) at 6% (*w/v*) and heated, first at 55°C for 30 min, second at 98 °C for 15 min, put on ice for 3 min and centrifuged for 5 min. The supernatant was transferred to a clean 1.5 ml microcentrifuge tube. The *bcrABC* and the *emrE* amplifications were done with primer pairs described by Elhanafi, Dutta [28] and Kovacevic, Ziegler [20], respectively, both using Q5 High-Fidelity 2X Master Mix (New England Biolabs) according to the manufacturer's instructions. Amplification products were analyzed on a 1.5% agarose gel.

### Benzalkonium chloride minimum inhibitory concentrations

*L. monocytogenes* resistance to BAC was confirmed phenotypically using a broth dilution assay to determine the minimum inhibitory concentrations (MIC) according to the Clinical and Laboratory Standards Institute [29] with minor modifications. Briefly, tryptic soy broth with yeast extract (TSBYE) supplemented with BAC (Sigma-Aldrich, Oakville, ON, Canada) at final concentrations ranging from 0 to 200 ppm (0, 0.78, 0.56, 3.12, 6.25, 12.5, 25, 50, 100 and 200) was inoculated with $10^5$ CFU of an overnight culture of *L. monocytogenes* using 96-well plates (Microtest Plate 96 well, Sarstedt, Montreal, QC, Canada). They were incubated at 30 °C for 48 h. The MIC was defined as the lowest concentration of BAC at which the growth of the bacteria was inhibited based on visual assessment. The assay was done in triplicate.

### Association between *L. monocytogenes* persistent pulsotypes and BAC resistance genes

The association between resistance to BAC following amplification of either *bcrABC* or *emrE* and confirmation by MIC, and the following variables 1) persistence of the pulsotype and 2) exposure of the site to disinfectants (low vs. high), were tested using an exact $\chi^2$ test, with the pulsotype as the unit of analysis.

## Results

### Distribution of *L. monocytogenes*-positive samples

A total of 2496 samples were collected from different operation areas over a 16-month period within four swine slaughterhouses (Table 1 and S1 Table) and assayed for the presence of *L. monocytogenes*. The bacterium was successfully isolated from 243 samples (9.7%). According to the final multivariable model, the proportion of *L. monocytogenes*-positive samples varied significantly according to slaughterhouses, operation areas and type of surface. Compared to slaughterhouse D, the detection of *L. monocytogenes* was significantly higher in slaughter-houses A and C (odds ratios (OR) = 1.9 and 3.6; $P = 0.05$ and $P < 0.001$, respectively; Table 1). No other statistically significant difference was observed between slaughterhouses. Compared to the slaughtering and bleeding area, the detection of *L. monocytogenes* was significantly higher in the dehairing and evisceration areas (OR = 5.7, $P < 0.001$), in the chilling and hanging area (OR = 5.5, $P < 0.01$), and in the cutting and deboning area (OR = 3.5; $P < 0.001$; Table 1). Finally, compared to the type of surface (sites, machines and materials), higher detection of *L. monocytogenes* was observed for the conveyor belt top side, the conveyor belt bottom

**Table 1. Distribution of *L. monocytogenes*-positive samples in four slaughterhouses in Québec, between October 2013 and February 2015.**

| Characteristics | Number of samples | Number (%) of *L. monocytogenes*- positive samples | Odds ratios[a] | |
|---|---|---|---|---|
| | | | Estimates | *P*-value[b] |
| **Seasons** | | | | |
| Fall | 624 | 44 (7.1) | Not included in the final model[c] | |
| Winter | 936 | 90 (9.6) | | |
| Spring | 468 | 52 (11.1) | | |
| Summer | 468 | 57 (12.2) | | |
| **Slaugherhouses** | | | | |
| A | 624 | 58 (9.3) | 1.9 | 0.05 |
| B | 624 | 55 (8.8) | 1.8 | 0.08 |
| C | 624 | 98 (15.7) | 3.6 | <0.001 |
| D | 624 | 32 (5.1) | Ref | |
| **Operation areas** | | | | |
| Lairage | 848 | 68 (8.0) | 1.5 | 0.55 |
| Slaughtering and bleeding | 288 | 10 (3.5) | Ref | |
| Dehairing and evisceration | 368 | 51 (13.9) | 5.7 | <0.001 |
| Chilling and hanging | 128 | 24 (18.8) | 5.5 | <0.01 |
| Cutting and deboning | 864 | 90 (10.4) | 3.5 | <0.001 |
| **Contact of surfaces with carcasses or meat** | | | | |
| Yes | 912 | 86 (9.4) | Not included in the final model[c] | |
| No | 1584 | 157 (9.9) | | |
| **Type of surface[d]** | | | | |
| Conveyor belt top side | 160 | 19 (11.9) | 2.7 | <0.01 |
| Conveyor belt bottom side | 160 | 21 (13.1) | 3.0 | <0.01 |
| Environment | 656 | 86 (13.1) | 3.0 | <0.001 |
| Machines | 256 | 27 (10.6) | 2.6 | <0.01 |
| Pens | 800 | 65 (8.1) | 4.0 | 0.04 |
| Materials | 464 | 25 (5.4) | Ref | |

Ref: Reference category used for odds ratios.

[a] Odds ratios from multi-level multivariable logistic regression model.

[b] *P*-value adjusted for multiple comparison (Tukey-Cramer).

[c] Rejected from the final multivariable model (*P*-value>0.05 during backward selection)

[d] Sites, machines and materials are described in S1 Table.

side, the environment, the machines, and the pens (OR = 2.7, 3.0, 3.0, 2.6 and 4.0, respectively; all $P \leq 0.05$; Table 1). The season and contact of surfaces with carcasses or meat were not significantly associated with *L. monocytogenes* positivity and thus were not kept in the final model.

## Characterization of *L. monocytogenes* isolates

Two unique isolates were obtained from 231 of the 243 *L. monocytogenes*-positive samples and one isolate from the remaining12 samples, for a total of 474 *L. monocytogenes* isolates. They were characterized by serotyping. The same serotype was observed for the two isolates in 95.2% (220) of the 231 samples. A sub-total of 217 samples (89.7%) contained at least one *L. monocytogenes* isolate from serotypes, 1/2a, 1/2b and/or 1/2c (Table 2).

**Table 2. Distribution of *L. monocytogenes*-positive samples according to serotypes detected in at least one isolate in four slaughterhouses in Québec.**

| Serotypes | Slaughterhouses | | | | | | | | | |
|---|---|---|---|---|---|---|---|---|---|---|
| | A(58 samples) | | B (55 samples) | | C (97 samples) | | D (32 samples) | | All (242 samples) | |
| | n | % | n | % | n | % | n | % | n | % |
| **1/2a** | 29 | 50.0 | 37 | 67.3 | 11 | 11.3 | 14 | 43.8 | 91 | 37.6 |
| **1/2b** | 4 | 6.9 | 8 | 14.6 | 56 | 57.7 | 12 | 37.5 | 80 | 33.1 |
| **1/2c** | 19 | 32.8 | 8 | 14.6 | 22 | 22.7 | 2 | 6.3 | 51 | 21.1 |
| **3a** | - | - | - | - | - | - | 5 | 15.6 | 5 | 2.1 |
| **3b** | - | - | - | - | 7 | 7.2 | - | - | 7 | 2.9 |
| **3c** | - | - | - | - | 1 | 1.0 | 1 | 3.1 | 2 | 0.8 |
| **4a** | 10 | 17.2 | - | - | - | - | - | - | 10 | 4.1 |
| **4b** | - | - | 3 | 5.5 | 4 | 4.1 | - | - | 7 | 2.9 |

n: number of *L. monocytogenes*-positive samples with the serotype detected in at least one isolate.

%: relative to the number of *L. monocytogenes*-positive samples in each slaughterhouse A, B, C or D, or in all slaughterhouses.

The 474 isolates were also characterized by ribotyping. A total of 18 ribotypes were conclusively identified and three additional ribotypes were tentatively identified. The ribotypes DUP-19165 (103 isolates from 57 samples), DUP-16619 (85 isolates from 46 samples), DUP-18627 (49 isolates from 26 samples) and DUP-1052 (45 isolates from 45 samples) were the most common confirmed ribotypes (S2 Table).

A sub-total of 321 isolates from 241 *L. monocytogenes*-positive samples were characterized by PFGE; one isolate was typed from 161 samples, two isolates having different ribotypes were typed from 69 samples, and two isolates sharing the same ribotype were typed from 11 samples. Overall, a total of 58 pulsotypes were revealed. Two different pulsotypes were detected in 10/69 samples with two isolates sharing a single ribotype, and 11/11 samples with two isolates having a different ribotype. Interestingly, 19 of these 58 pulsotypes, were novel additions in the PulseNet Canada database (Fig 1). Seven pulsotypes were detected in at least one *L. monocytogenes* isolate from 10 different samples (Fig 1). Pulsotypes LS4 and LS24 were the most common and included the highest number of *L. monocytogenes* isolates, 34 (from 28 samples) and 33 (from 24 samples), respectively.

A dendrogram of the PFGE profiles based on ApaI and AscI restriction enzymes from the 58 pulsotypes is shown in Fig 2A and 2B, respectively. Three lineages, I, II and III, were identified. Lineages I and II include 24 and 33 pulsotypes, respectively. Lineage III includes a single pulsotype. Serotypes 1/2a and 1/2b are predominantly present and comprise 18 and 17 pulsotypes, distributed mainly in lineages II and I, respectively. Serotype 1/2c comprises 13 pulsotypes, all in lineage II. Serotypes 3a, 3b, 4a and 4b comprise 2, 3, 1 and 4 pulsotypes, respectively. The single pulsotype in lineage III is the sole member of serotype 4a.

## Distribution of *L. monocytogenes* pulsotypes according to the slaughterhouse

The distribution of the 58 pulsotypes varied according to the slaughterhouse. A total of 33 different pulsotypes were isolated from slaughterhouse C. Eleven, 13 and 13 different pulsotypes were isolated from slaughterhouses A, B and D, respectively (Fig 3). Ten pulsotypes were present in more than one slaughterhouse. A single pulsotype, LS4 (serotype 1/2a), was present in all four slaughterhouses. Pulsotypes LS12 (1/2c) and LS24 (1/2c) were only present in

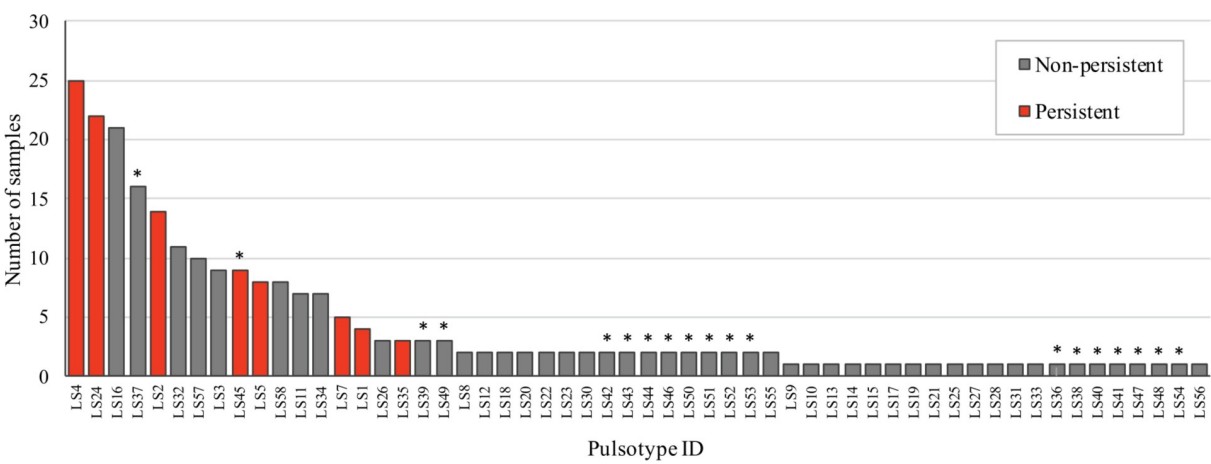

**Fig 1. Representation of the number of *L. monocytogenes*-positive samples according to their pulsotype.** The persistent pulsotypes are colored in red. The newly discovered pulsotypes (absent from the PulseNet Canada database prior to our study) are labeled with an asterisk.

slaughterhouses A and B; LS2 (1/2a) only in A and C, LS45 (1/2b) in A and D, LS37 (1/2b) in B and C, and LS47(3c), LS50 (1/2b), LS51(1/2a) and LS58 (1/2c) in C and D (Fig 3).

**Fig 2.** Dendrogram of PFGE-ApaI (A) and AscI (B) profiles from 58 *L. monocytogenes* pulsotypes.

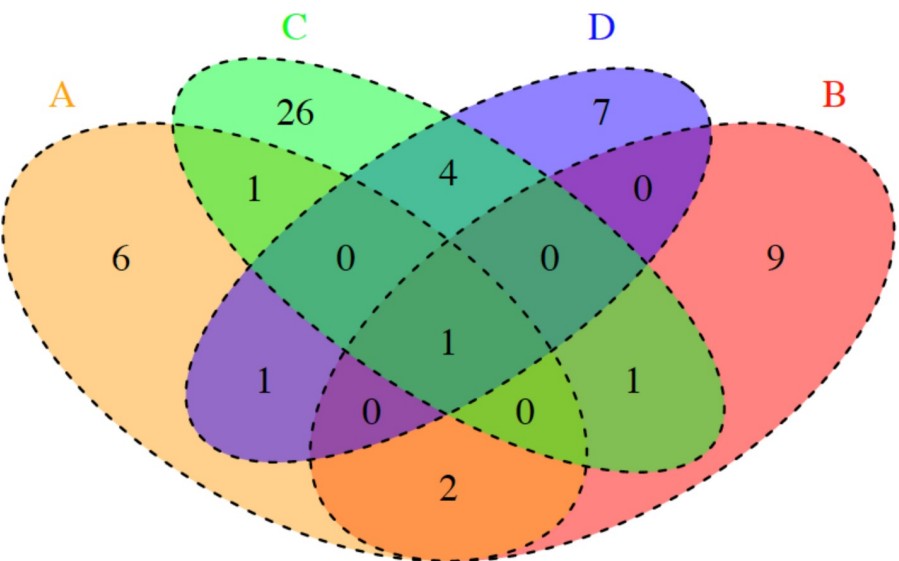

**Fig 3. Venn diagram with the numbers of shared pulsotypes among the four slaughterhouses A, B, C and D.**

## Diversity of *L. monocytogenes* pulsotypes according to seasons, slaughterhouses and operation areas

According to the rarefaction curves, the *L. monocytogenes* pulsotype diversity was the lowest in fall (Fig 4A) and was significantly lower in fall than in summer (Fig 4B, S1 Fig). The *L. monocytogenes* pulsotype diversity was significantly higher in slaughterhouse C compared to A and B (Fig 4C and 4D). No significant differences were observed in the diversity between operation areas.

## Distribution of *L. monocytogenes* persistent pulsotypes

Eight pulsotypes, (LS1, LS2, LS4, LS5, LS7, LS25, LS35 and LS45) were defined as persistent. The proportion of persistent pulsotypes was statistically similar in all slaughterhouses except for C (Table 3). The OR of contamination with a persistent pulsotype was significantly higher in slaughterhouses A, B and D compared to slaughterhouse C (OR = 14.4, 5.4 and 9.6 respectively; all $P < 0.01$). In the disinfected operation areas of the slaughterhouse, *i.e.* dehairing and evisceration, chilling and hanging, and cutting and deboning areas, the OR of sample contamination with persistent pulsotype was significantly higher compared to the lairage operation area (OR = 6.7, 7.5 and 36.5 with $P = 0.02$, $P = 0.05$ and $P < 0.001$, respectively; Table 3). Moreover, the OR was higher in the cutting and deboning areas compared to the dehairing and evisceration areas (OR = 4.89, $P < 0.01$).

## Association between *L. monocytogenes* persistent pulsotypes and detection in food and clinical human listeriosis cases

Of the 58 different *L. monocytogenes* pulsotypes isolated in our study, 11 were detected at least once in food and/or human listeriosis cases. A sub-total of six pulsotypes (LS7, LS11, LS14, LS19, LS24 and LS28) were isolated only from food, one (LS31) was isolated only from human listeriosis cases, and four (LS1, LS2, LS4 and LS57) from both food and human listeriosis cases. The 19 newly identified pulsotypes were not included in this analysis since it was not possible to match them with pulsotypes in the LSPQ database of food and human listeriosis cases.

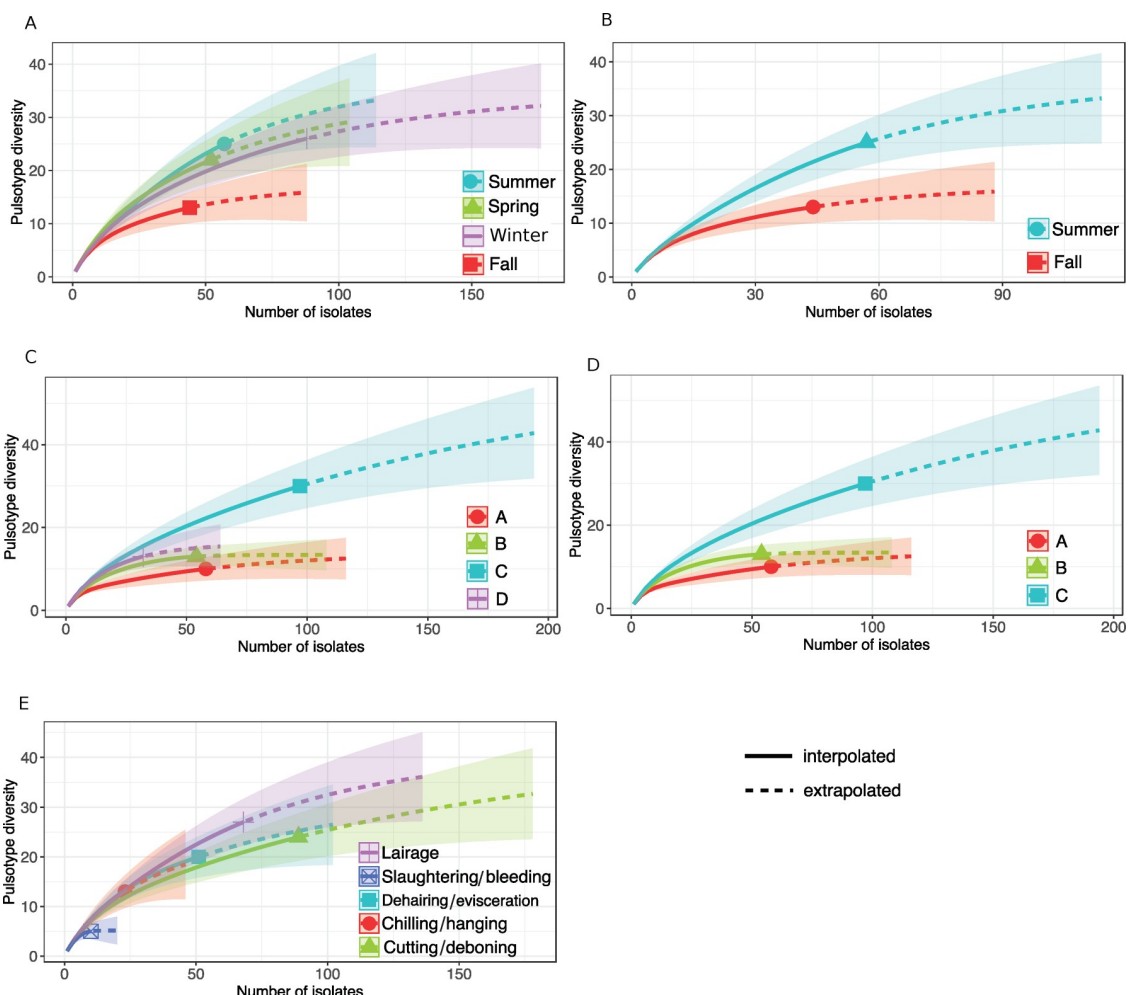

**Fig 4. Pulsotype diversity over the number of *L. monocytogenes* isolates collected from all four swine slaughterhouses.** A) according to the four seasons. B) according to two seasons, summer and fall. C) according to the four slaughterhouses, A, B, C and D. D) according to the slaughterhouses A, B, C. E) according to five operation areas. One characterized isolate per sample was selected randomly.

Statistically significant associations between *L. monocytogenes* persistent pulsotypes and their detections in food or human listeriosis cases were revealed. From the ten pulsotypes observed in food, five (50%) were persistent whereas only two (6.9%) from the pulsotypes not observed in food were persistent (Fig 5, $P < 0.01$, exact $\chi^2$). Similarly, from the five pulsotypes observed in human listeriosis cases, three (60%) were persistent whereas only four (11.8%) from the 34 pulsotypes not observed in human listeriosis cases were persistent (Fig 5, $P < 0.01$, exact $\chi^2$).

## Association between *L. monocytogenes* persistent pulsotypes and BAC resistance genes

The presence of BAC resistance genes was strongly associated with *L. monocytogenes* pulsotypes designated persistent ($P < 0.01$, exact $\chi^2$). Among the 14 pulsotypes harboring the BAC resistance genes, six (42.8%) were persistent. Comparatively, only two (4.5%) from the 44

**Table 3. Distribution of *L. monocytogenes*-positive samples and persistent pulsotypes in four slaughterhouses in Québec.**

| Characteristics | Number of samples[a] | Number (%) with persistent strains | Odds ratios[b] | |
|---|---|---|---|---|
| | | | Estimate | *P*-value[c] |
| **Slaughterhouses** | | | | |
| A | 58 | 33 (56.9%) | 14.4 | <0.001 |
| B | 54 | 32 (59.3%) | 5.4 | <0.01 |
| C | 97 | 12 (12.4%) | Ref | |
| D | 32 | 13 (40.6%) | 9.6 | <0.01 |
| **Operation areas** | | | | |
| Lairage | 68 | 5 (7.4%) | Ref. | |
| Slaughtering and bleeding | 10 | 3 (30.0%) | 9.0 | 0.15 |
| Dehairing and evisceration | 51 | 13 (25.5%) | 6.7 | 0.02 |
| Chilling and hanging | 23 | 8 (34.8%) | 7.5 | 0.05 |
| Cutting and deboning | 89 | 61 (68.5%) | 36.5 | <0.001 |

Ref.: Reference category used for odds ratios.

[a] Only *L. monocytogenes*-positive samples from the 241 samples with at least one isolate characterized by PFGE were included (two samples were missing). One isolate per sample was randomly selected.

[b] Odds ratios from multi-level multivariable logistic regression model.

[c] *P*-value adjusted for multiple comparison (Tukey-Cramer).

pulsotypes without the BAC resistance genes were persistent (Fig 6). Interestingly, the *emr E* gene was found only in *L. monocytogenes* pulsotype LS1 belonging to the clonal complex 8 (CC8).

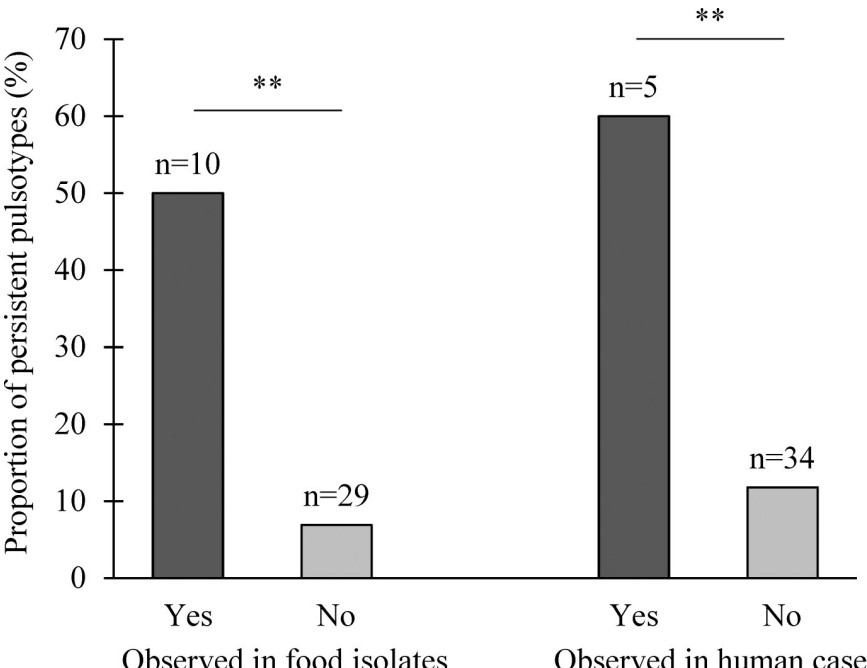

**Fig 5. Proportion of persistent *L. monocytogenes* pulsotypes observed in food isolates and human listeriosis cases from 2000 to 2016 in Quebec according to the database of the *Laboratoire de santé publique du Québec* (LSPQ).** [**]: *P* < 0.01.

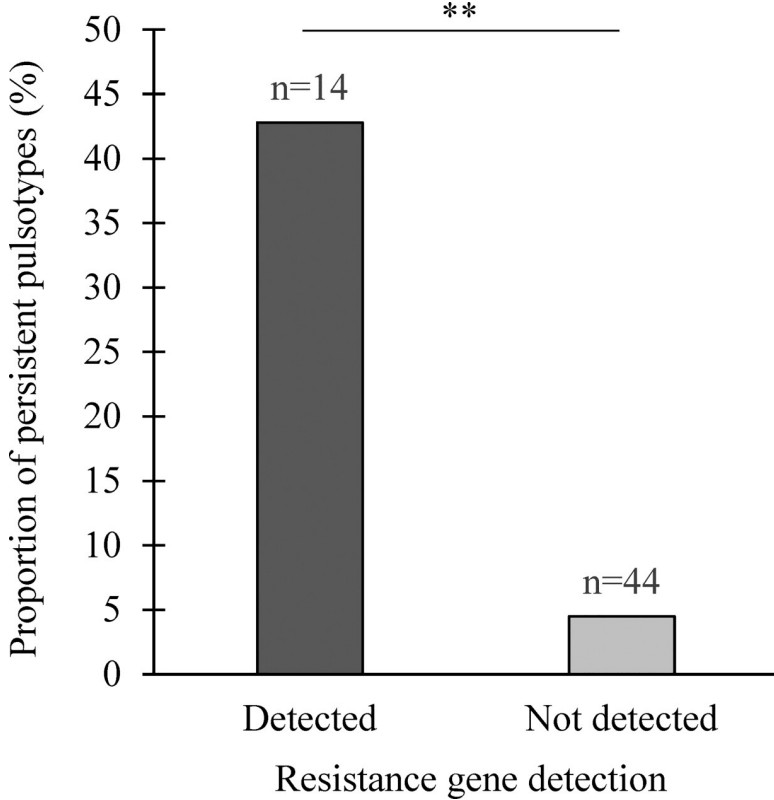

**Fig 6. Proportion of persistent *L. monocytogenes* pulsotypes harboring the benzalkonium chloride *bcrABC* resistance cassette and/or the *emrE* multidrug transporter gene.** **: $P < 0.01$.

The MICs of *L. monocytogenes* isolates harboring *bcrABC* or *emrE* (n = 14) were 6.2 ppm and 3.2 ppm, respectively. Conversely, the MICs of *L. monocytogenes* isolates lacking both *bcrABC* and *emrE* were two- to eight-fold lower.

The presence of BAC resistance genes in *L. monocytogenes* pulsotypes was tested for its association to the level of site exposure to disinfectants. No significant association was revealed ($P = 0.8$, exact $\chi^2$; Fig 7).

## Discussion

*L. monocytogenes* isolate diversity has been studied in the pork meat production chain in European countries [7, 27, 30–34], However, to the best of our knowledge, our study is the first extensive survey of the distribution, diversity and persistence of *L. monocytogenes* isolates in the various operation areas in swine slaughterhouses over a 16-month period. The overall proportions of *L. monocytogenes*-positive samples in slaughterhouses was in agreement with those previously reported European countries [5, 27] when the sampling was done following cleaning and sanitation (Table 1). The detection frequency of *L. monocytogenes* on surfaces before cleaning and sanitation seems to be higher and was reported to reach up to 33% [32, 33]. In contrast, the probability of contamination of surfaces observed in our study was relatively low (9.7%, Table 1) as compared with the 24% reported for raw materials and pork products [27] which could be explained by re-contamination during the post-slaughtering operation areas [6, 35, 36].

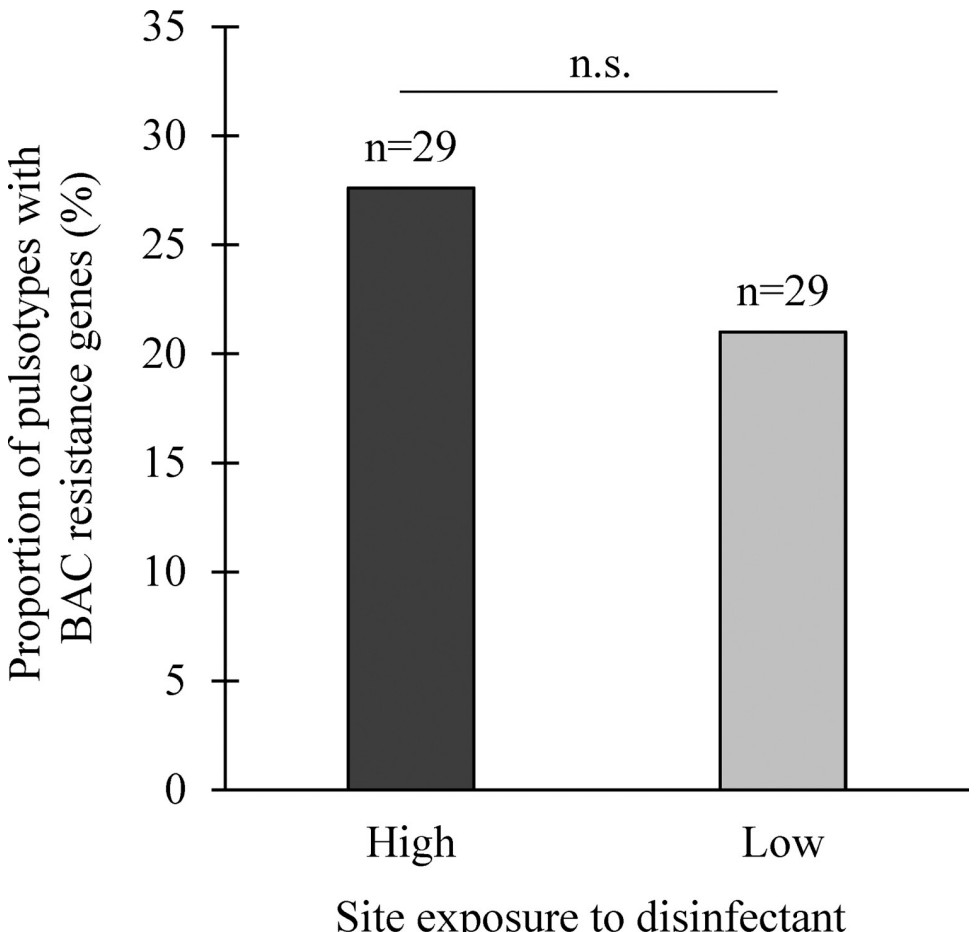

**Fig 7. Proportion of *L. monocytogenes* pulsotypes harboring the benzalkonium chloride *bcrABC* resistance cassette and/or the *emrE* multidrug transporter gene relative to their origin, the most (yes) vs. the least exposed (No) sites to disinfectants.** N.S: *P* > 0.05.

We observed a higher level of surface contamination with *L. monocytogenes* in plant C compared to slaughterhouse D (Table 1). One reason could be related to the export activity of the slaughterhouse D where the implemented pathogen surveillance is more rigorous to meet export requirements. The dehairing and evisceration, chilling and hanging and cutting and deboning areas were the more likely contaminated by *L. monocytogenes* compared to the lairage and slaughtering areas (Table 1). These results could be explained by meat product acting as a source of environmental contamination, considering the high probability of meat raw products contamination reported in previous studies [4, 5, 27]. The practice of evisceration could increase the environmental contamination risk given the presence of *L. monocytogenes* in many parts of a pig, such as the tongue and tonsils. Indeed, the prevalence of this pathogen in the evisceration step can be as high as 24% [27, 37]. Moreover, a competition factor should be taken into consideration in the chilling and hanging and the cutting and deboning areas, where contamination by mesophilic bacteria could be at its lowest numbers in favor of the psychrophilic *L. monocytogenes* growth [38, 39]. Previous studies have proposed that the chilling and cutting rooms could be the most likely source of raw product contamination due to their high probability of *L. monocytogenes* contamination [6].

Serotypes 1/2a and 1/2b were the most frequently *L. monocytogenes* serotypes observed (Table 2). This is in agreement with previous studies conducted following cleaning and sanitation procedures [4, 27, 40–42]. Conversely, in a recent study by Meloni *et al.*, the serotype 1/2c was the most frequently observed [32]. However, because the sampling was done before the cleaning and sanitation procedures [32], it is tempting to speculate that the latter affects the survival of the serotypes. Our results suggest that serotype 1/2a and 1/2b would better survive the cleaning and sanitation procedures.

A total of 58 different pulsotypes were identified by PFGE typing (Fig 1). Here, only one isolate per sample was typed for most samples harboring isolates with identical ribotypes. Considering that 14.5% of the 69 randomly selected samples with identical ribotypes had different pulsotypes, the presence of certain pulsotypes could have been missed. The selection of one isolate by sample could have resulted in an underestimation of the pulsotype diversity and/or of the proportion of samples with persistent strains. However, this should not have biased our results on the comparison of diversity and/or strain persistence between slaughterhouses, seasons or operation areas, as only one randomly selected isolate per sample was kept in these analyses to ensure group comparability.

In our study, the diversity of *L. monocytogenes* strains appears to be season-dependent. It was higher in summer than in fall. It is difficult to pinpoint the factors behind this difference. On the other hand, however, the non-significant differences in the detection level of *L. monocytogenes* between seasons suggests that strains diversity would not be dependent on prevalence.

Slaughterhouse C showed a higher *L. monocytogenes* diversity compared to slaughterhouses A and B (Fig 4). Interestingly, this slaughterhouse had a higher production capacity. The three other slaughterhouses, A, B and D, were supplied only by local farms (S. Fournaise, personal communication). Interestingly, the high prevalence of L. *monocytogenes* in the slaughterhouse C appears to be correlated to the higher diversity of pulsotypes found in this plant (S1 Fig).

Surprisingly, the proportion of persistent strains among *L. monocytogenes*-positive samples was lower in slaughterhouse C (Table 3). One explanation could be related to a less efficient cleaning and disinfection in this slaughterhouse, therefore reducing the selective pressure on persistent strains. At the operation areas following slaughtering and bleeding, the increasing proportion of isolates from persistent strains over the slaughtering process was noticeable (Table 3). This could be explained by differences in the disinfection protocols. Indeed, the lairage and slaughtering and bleeding areas were in proximity and had similar cleaning and sanitation procedures. They were subjected to weekly disinfections using alcohol (70; % *v/v*) and potassium monopersulfate (1%; *m/v*)-based disinfectants. In the other operation areas (i.e. dehairing and evisceration, chilling and hanging, cutting and deboning), the disinfection was done daily using QACs (from 150 to 200 ppm), thus, *L. monocytogenes* strains were subjected to the same environmental conditions. Furthermore, the emergence of resistance or tolerance to QAC could be favored in the presence of sub-optimal concentrations of the disinfectant on localized wet surfaces due to a dilution effect, as reported in many studies [14, 17, 43, 44]. Furthermore, complex equipment and machines present in the dehairing and evisceration, cutting and deboning areas could favor the establishment of harborage sites, which are often present in food equipment and premises [45, 46].

Because of the importance of *L. monocytogenes* persistent strains in food contamination and, as a consequence to the risk for human health, we aimed to study whether the persistent profiles were more likely to be detected in food and human listeriosis strains. We observed a significant association (Fig 5), which may indicate a high presence of these persistent profiles throughout Quebec, leading to repeated contamination of food and, more importantly, suggesting that these pulsotypes have been present in food and/or food processing environments

over the last decade. Interestingly, the S2 and S4 pulsotypes are frequently isolated from sporadic cases of listeriosis (S. Bekal, unpublished data) and the persistent S1 pulsotype belonging to the CC8 is the predominant clone that has been present in listeriosis outbreaks in Canada for more than two decades [47]. A previous study showed that one persistent *L. monocytogenes* strain isolated over a 12-year period in a food processing facility was associated with repeated listeriosis outbreaks [11]. Nonetheless, our study design does not allow us to determine if the strains isolated from human listeriosis cases originated from the slaughterhouses included in our study.

The significant association between specific *L. monocytogenes* pulsotype persistence and resistance to BAC, as confirmed by MIC, provides important insights about how these persistent strains can survive in the slaughterhouse over a long period. In the present study, we showed that either the *bcrABC* resistance cassette or the *emrE* gene, or both, were present in persistent *L. monocytogenes* strains. In other studies, BAC resistance genes have been reported in *L. monocytogenes* strains from food processing facilities, with different prevalences in various region in the world [14, 44, 48–50]. However, only a few studies have shown an association between persistence of *L. monocytogenes* strains and the presence of resistant genes to BAC [43, 44]. In this study, we observed that the proportion of *L. monocytogenes* strains with resistant genes to BAC was significantly higher in persistent strains than in non-persistent ones, suggesting that the resistance to BAC may be an important contributor to the persistence of *L. monocytogenes* strains in the studied slaughterhouses.

## Conclusion

This study provides for the first report on the presence, distribution and persistence of *L. monocytogenes* strains in Canadian swine slaughterhouses. Our study found a wide diversity of *L. monocytogenes* isolates in slaughterhouses, indicative of their ability to survive and grow within different environment conditions. Furthermore, the high prevalence of persistent *L. monocytogenes* strains in the slaughterhouse cutting and deboning operation area should be taken into consideration. Meat contamination by persistent strains in the operation area could be carried over to the downstream pork meat processing facilities and would pose a risk to human health.

## Supporting information

**S1 Fig. Pulsotype diversity of *L. monocytogenes* isolated from four swine slaughterhouses in Quebec, Canada.** A. Pulsotype diversity according to the seasons (A: Fall, B: Winter, C: Spring, D: Summer). B. Pulsotype diversity according to slaughterhouses (A, B, C, D). C. Pulsotype diversity according to the operation areas in the slaughterhouse (A: Lairage, B: Slaughtering and bleeding, C: Dehairing and evisceration, D: Chilling and hanging, E: Cutting and deboning)
(PDF)

**S1 Table. Sampling of four swine slaughterhouses in Quebec, Canada, for the isolation of *L. monocytogenes*.**
(PDF)

**S2 Table. Characteristics of the sample of origin, typing results, and resistance to QACs of the 474 *L. monocytogenes* isolates.**
(XLSX)

## Acknowledgments

We thank Kevin Tyler and Karine Hébert from Health Canada for strain characterization.

## Author Contributions

**Conceptualization:** Tamazight Cherifi, Julie Arsenault, Franco Pagotto, Sylvain Quessy, Philippe Fravalo.

**Data curation:** Tamazight Cherifi, Franco Pagotto, Jean-Charles Côté, Kersti Neira, Sadjia Bekal, Philippe Fravalo.

**Formal analysis:** Tamazight Cherifi, Julie Arsenault, Franco Pagotto, Jean-Charles Côté, Sadjia Bekal.

**Funding acquisition:** Sylvain Quessy, Philippe Fravalo.

**Investigation:** Franco Pagotto, Philippe Fravalo.

**Methodology:** Tamazight Cherifi, Julie Arsenault, Sylvain Quessy, Jean-Charles Côté, Kersti Neira, Sylvain Fournaise, Sadjia Bekal, Philippe Fravalo.

**Project administration:** Philippe Fravalo.

**Resources:** Julie Arsenault, Franco Pagotto, Sylvain Quessy, Sylvain Fournaise, Philippe Fravalo.

**Software:** Tamazight Cherifi, Julie Arsenault, Franco Pagotto, Philippe Fravalo.

**Supervision:** Sylvain Quessy, Philippe Fravalo.

**Validation:** Tamazight Cherifi, Julie Arsenault, Sylvain Quessy, Jean-Charles Côté, Philippe Fravalo.

**Visualization:** Tamazight Cherifi, Julie Arsenault, Jean-Charles Côté, Kersti Neira.

**Writing – original draft:** Tamazight Cherifi, Julie Arsenault, Franco Pagotto, Sylvain Quessy, Jean-Charles Côté, Kersti Neira, Sylvain Fournaise, Sadjia Bekal, Philippe Fravalo.

**Writing – review & editing:** Tamazight Cherifi, Julie Arsenault, Franco Pagotto, Jean-Charles Côté, Philippe Fravalo.

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
