## [Decision Letter · Decision Letter 0]

18 Jun 2020

PONE-D-20-16869

Distribution, diversity and persistence of Listeria monocytogenes in swine slaughterhouses and their association with food and human listeriosis strains

PLOS ONE

Dear Dr.Cherifi,

Thank you for submitting your manuscript to PLOS ONE. After careful consideration, we feel that it has merit but does not fully meet PLOS ONE’s publication criteria as it currently stands. Therefore, we invite you to submit a revised version of the manuscript that addresses the points raised during the review process.

Your manuscript has been reviewed by two experts in your filed.  A minor revision is suggested.  

We look forward to receiving your revised manuscript.

Kind regards,

Yung-Fu Chang

Academic Editor

PLOS ONE

Journal Requirements:

2. In the interests of reproducibility, in your Methods, please state the names/locations of the slaughterhouses where samples were collected. If you are unable to share this information due to privacy agreements, please include an explanation in your Ethics Statement.

I have read the journal's policy and the authors of this manuscript have the following

competing interests:Authors declare no competing of interest. Dr Sylvain Fournaise

(from Olymel) contributed to the sampling design at the slaughterhouses and the

interpretation of results, but had no role in laboratory analyses, statistical analyses or in

reporting the results presented in this work.

Reviewers' comments:

Reviewer's Responses to Questions

**Comments to the Author**

1. Is the manuscript technically sound, and do the data support the conclusions?

Reviewer #1: Yes

Reviewer #2: Partly

2. Has the statistical analysis been performed appropriately and rigorously? 

Reviewer #1: Yes

Reviewer #2: Yes

3. Have the authors made all data underlying the findings in their manuscript fully available?

Reviewer #1: Yes

Reviewer #2: Yes

4. Is the manuscript presented in an intelligible fashion and written in standard English?

Reviewer #1: Yes

Reviewer #2: Yes

5. Review Comments to the Author

Reviewer #1: Manuscript Number: PONE-D-20-16869

In this manuscript, the authors provided a deep analysis of L. monocytogenes distribution in the pork production chain from Quebec, Canada, demonstrating risk factors and relevance of production aspects on the presence of persistent strains, and also correlated with isolates previously obtained from other samples. Minor review is suggested:

Please, provide results to support your comments. like in lines 33-34: how these proportions varied based on slaughterhouses and operation areas?

Line 35-37: please, delete “..according to our definition (i.e isolated at least three times in the same slaughterhouse over the 16-month period, with a four- to five-month interval between each sampling visit)”

Line 38: indicate p value for the significant finding

Line 40: how much these genes were more frequent in persistent strains?

Line 40-42: I would expect a much more interesting conclusion than this… this is already known… maybe you can add information related to resistance strains in this conclusion, a real important result.

Line 50: replace Listeria by L.

Line 62-65: this information is important! The authors need to provide further information about resistance of L. monocytogenes to sanitizers and how this is relevant to result in strain persistence. Then, you can address information about quaternary ammonium compounds and Listeria resistance.

Line 66-69: this information could be better merged with the paragraph 53-61… in the present format is loose

Line 102: replace Listeria by L.

Line 112-113: please, indicate the target gene considered to confirm the isolates as L. monocytogenes

Line 187: replace Listeria by L.

Please, consider presenting the digestion profiles in figures, to allow the reader to visualize them and check their quality

Discussion: please, indicate the figures and tables with the obtained results to support your discussion.

Reviewer #2: The manuscript PONE-D-20-16869 “Distribution, diversity and persistence of Listeria monocytogenes in swine slaughterhouses and their association with food and human listeriosis strains” represents a very good application of typing methods and statistical logistic regression for the study of L. monocytogenes dissemination within slaughterhouses. It is also very interesting in discussing the persistency in relation to sample sources, type and presence of disinfectant resistance. However, I suggest some improvements below:

Abstract: need specification on statistical methods used for assessing association between persistency and virulence/resistance

Introduction:

Line 55-57 need clarification

Materials and Methods:

Line 97: what do you mean by “put on ice”. Please, clarify

Line 123-125: please clarify

Line 158-163: redundancy with the previously presented statistical methods

Line 195: “visually observed” needs clarification

Results:

Line 233-235: try to clarify the sentence to make numbers clearer.

Line 242: check the table notes

Line 302-304: no need, already stated in M and M

Discussion:

Line 372: check was/is

Line 376-381: clarify sentence and explain better what do you mean by “subjectively observed …” what was observed to conclude that in one premise the cleaning practices were more effective? Check lists? Observation of some particular aspects?

Line 396-399 need some more support, maybe a table, if not is just conjecture

Line 406-412: too much speculation

Line 415: ok, but you need to discuss more in depth the meaning of the underestimation of variability. Could it impair your results or conclusion?

Line 432-437: need more explanation on the differences in cleaning Line 438-439: speculation (did you see these evidence during your visits?)

Line 464-467: ok but mitigate: you have very few samples studied to say that you demonstrated something

Line 468-470: speculation

6. PLOS authors have the option to publish the peer review history of their article (what does this mean?). If published, this will include your full peer review and any attached files.

Reviewer #1: No

Reviewer #2: No

---

## [Author Response · Author response to Decision Letter 0]

10 Jul 2020

Response to Reviewer’s Comments - Manuscript PLOS D 20 16869

We would like to thank the reviewers for their time and constructive comments, which definitely help to improve the quality of our manuscript. 

Reviewer #1 

In this manuscript, the authors provided a deep analysis of L. monocytogenes distribution in the pork production chain from Quebec, Canada, demonstrating risk factors and relevance of production aspects on the presence of persistent strains, and also correlated with isolates previously obtained from other samples. Minor review is suggested:

1. Please, provide results to support your comments. like in lines 33-34: how these proportions varied based on slaughterhouses and operation areas? 

Response: The range in proportions for slaughterhouses and operation areas were added in the Abstract (lines 34-35). 

2. Line 35-37: please, delete “..according to our definition (i.e isolated at least three times in the same slaughterhouse over the 16-month period, with a four- to five-month interval between each sampling visit)”

Response: This part of the sentence was deleted as suggested. 

3. Line 38: indicate p value for the significant finding. 

Response: P-values were added for each of the two comparisons (lines 38-40). 

4. Line 40: how much these genes were more frequent in persistent strains?

Response: We modified the sentences in the Abstract to better present this association, including the frequency in each group (lines 39-41). 

5. Line 40-42: I would expect a much more interesting conclusion than this… this is already known… maybe you can add information related to resistance strains in this conclusion, a real important result. 

Response: We improved the conclusion in the Abstract (lines41-44).

6. Line 50: replace Listeria by L. 

Response: The correction was done. We performed a full-text search and corrected another occurrence of the same error.

7. Line 62-65: this information is important! The authors need to provide further information about resistance of L. monocytogenes to sanitizers and how this is relevant to result in strain persistence. Then, you can address information about quaternary ammonium compounds and Listeria resistance.

Response: We added more information according to the literature (lines 61-71).

8. Line 66-69: this information could be better merged with the paragraph 53-61… in the present format is loose. 

Response: True. We merged the two paragraphs in order to be more coherent (lines 61-71). 

9. Line 102: replace Listeria by L. 

Response: Correction was provided within the text.

10. Line 112-113: please, indicate the target gene considered to confirm the isolates as L. monocytogenes. 

Response: Target genes are now listed in the text (prfA, prs, lmo0737, lmo1118, orf2819, orf2110, and flaA genes) (line 120).

11. Line 187: replace Listeria by L. 

Response: Correction was provided within the text.

12. Please, consider presenting the digestion profiles in figures, to allow the reader to visualize them and check their quality.

Response: Thank you for your comment. We have added the gel images to the dendrogram in Figures 2A and 2B.

13. Discussion: please, indicate the figures and tables with the obtained results to support your discussion.

Response: We have added references to the Tables and Figures throughout the Discussion, as suggested. 

Reviewer #2

Reviewer #2: The manuscript PONE-D-20-16869 “Distribution, diversity and persistence of Listeria monocytogenes in swine slaughterhouses and their association with food and human listeriosis strains” represents a very good application of typing methods and statistical logistic regression for the study of L. monocytogenes dissemination within slaughterhouses. It is also very interesting in discussing the persistency in relation to sample sources, type and presence of disinfectant resistance. However, I suggest some improvements below:

1. Abstract: need specification on statistical methods used for assessing association between persistency and virulence/resistance

Response: An exact chi-square test was used for these analyses; this precision was added in the Abstract as suggested (lines 38, 41) .

2. Introduction: Line 55-57 need clarification. 

Response: Agreed. Some text was added to improve clarification (lines 55-57).

3. Materials and Methods: Line 97: what do you mean by “put on ice”. Please, clarify. 

Response: ”put on ice” has been replaced by “… kept in a cooler with ice packs…” (line 103).

4. Line 123-125: please clarify. 

Response: This sentence now reads as “One isolate per sample and per ribotype was selected for PFGE typing (i.e. if two isolates from a same sample had different ribotype, both were selected for PFGE typing; otherwise, one isolate was randomly selected). In addition, for a subset of randomly selected samples with two isolates sharing the same ribotype, the two isolates were typed.” We hope that the sentence is now clearer.

5. Line 158-163: redundancy with the previously presented statistical methods.

Response: We replace “(…) with the visit included as a random effect. The GLIMMIX procedure of SAS v 9.4 (SAS Institute Inc., Cary, NC, USA) was used with Laplace estimation. Odds ratios were used to present results, with P-values adjusted for multiples comparisons using Tukey-Cramer tests” by “as described above or the analysis of L. monocytogenes-positive samples. “ (lines 174-175).

6. Line 195: “visually observed” needs clarification. 

Response: The sentence now reads as: “The MIC was defined as the lowest concentration of BAC at which the growth of the bacteria was inhibited based on visual assessment “. (lines 211-212)

7. Results: Line 233-235: try to clarify the sentence to make numbers clearer. 

Response: We understand that the direct reference to odds ratios can lead to cumbersome sentences. We rephrase them to put emphasis on the direction of the associations. (lines 281-292)

8. Line 242: check the table notes. 

Response: The superscript was removed when referring to an abbreviation and kept otherwise (all Tables were revised accordingly).

9. Line 302-304: no need, already stated in M and M

Response: Agreed. This sentence was removed.

10. Discussion: Line 372: check was/is. 

Response: “is” was deleted. 

11. Line 376-381: clarify sentence and explain better what do you mean by “subjectively observed …” what was observed to conclude that in one premise the cleaning practices were more effective? Check lists? Observation of some particular aspects?

Response: Thank you for your comment. Because we had no additional information or data to confirm our observations, we chose to delete these sentences.

12. Line 396-399 need some more support, maybe a table, if not is just conjecture

Response: Agreed. It was conjecture. We have chosen to delete this paragraph because we had no information to support these hypotheses.

13. Line 406-412: too much speculation

Response: We agree. Thank you for raising the point. These sentences were removed as they could lead to over-interpretation.

14. Line 415: ok, but you need to discuss more in depth the meaning of the underestimation of variability. Could it impair your results or conclusion?

Response: We added the following information in this paragraph: “The selection of one isolate by sample could have resulted in an underestimation of the pulsotype diversity and/or of the proportion of samples with persistent strains. However, this should not have biased our results related to the comparison of diversity and/or strain persistence between slaughterhouses, seasons or operation areas, as only one randomly selected isolate per sample was kept for theses analyses to ensure group comparability”.

15. Line 432-437: need more explanation on the differences in cleaning. 

Response: We added the following explanation in these differences:” The lairage and slaughtering and bleeding areas were located closed to each other and had similar cleaning and sanitation procedures. They were subjected to a weekly disinfection using alcohol (70 %; v/v) and potassium monopersulfate (1 %; m/v) based disinfectants. In the other operation areas (i.e. dehairing and evisceration, chilling and hanging, cutting and deboning), the disinfection was done daily using QACs (from 150 to 200 ppm), thus, L. monocytogenes strains were subjected to the same environmental conditions”.

16. Line 438-439: speculation (did you see these evidence during your visits?):

Response: In the slaughterhouses we sampled, but also in food processing plants in general, the surfaces are always wet due to the daily cleaning and sanitation. This can lead to a dilution of the disinfectants in micro-places within the slaughterhouses or the food processing plants.

17. Line 464-467: ok but mitigate: you have very few samples studied to say that you demonstrated something

Response: We substituted “demonstrated” by “observed”, and modified slightly the rest of the sentence to tone down our conclusion here 

18. Line 468-470: speculation

Response: We agree. Thank you for your comment. This sentence was removed.

---

## [Editor Report · Decision Letter 1]

15 Jul 2020

Distribution, diversity and persistence of Listeria monocytogenes in swine slaughterhouses and their association with food and human listeriosis strains

PONE-D-20-16869R1

Dear Dr. Cherifi,

We’re pleased to inform you that your manuscript has been judged scientifically suitable for publication and will be formally accepted for publication once it meets all outstanding technical requirements.

Kind regards,

Yung-Fu Chang

Academic Editor

PLOS ONE
---

## [Editor Report · Acceptance letter]

23 Jul 2020

PONE-D-20-16869R1 

Distribution, diversity and persistence of Listeria monocytogenes in swine slaughterhouses and their association with food and human listeriosis strains 

Dear Dr. Cherifi:

I'm pleased to inform you that your manuscript has been deemed suitable for publication in PLOS ONE. Congratulations! Your manuscript is now with our production department. 

Kind regards, 

on behalf of

Dr. Yung-Fu Chang 

Academic Editor

PLOS ONE